# Distributionally Robust Learning for Uncertainty Calibration under Domain Shift

## Abstract

We propose a framework for learning calibrated uncertainties under domain shifts. We consider the case where the source (training) distribution differs from the target (test) distribution. We detect such domain shifts through the use of binary domain classifier and integrate it with the task network and train them jointly end-to-end. The binary domain classifier yields a density ratio that reflects the closeness of a target (test) sample to the source (training) distribution. We employ it to adjust the uncertainty of prediction in the task network. This idea of using the density ratio is based on the distributionally robust learning (DRL) framework, which accounts for the domain shift through adversarial risk minimization. We demonstrate that our method generates calibrated uncertainties that benefit many downstream tasks, such as unsupervised domain adaptation (UDA) and semi-supervised learning (SSL). In these tasks, methods like self-training and FixMatch use uncertainties to select confident pseudo-labels for re-training. Our experiments show that the introduction of DRL leads to significant improvements in cross-domain performance. We also demonstrate that the estimated density ratios show agreement with the human selection frequencies, suggesting a positive correlation with a proxy of human perceived uncertainties.

## 1 Introduction

Uncertainty estimation is an important machine learning problem that is central to trustworthy AI (Tomsett et al., 2020; Antifakos et al., 2005). In addition, many important downstream applications rely on the correct estimation of uncertainties. This includes unsupervised domain adaptation (UDA) (Zou et al., 2019) and semi-supervised learning (SSL) (Sohn et al., 2020) where they are used to solicit confident pseudo-labels for re-training. In these applications, reliable pseudo-labels help avoid error propagation and catastrophic failures in early iterations (Kumar et al., 2020).

Obtaining reliable uncertainty estimation is challenging. In contrast to human annotation of labels, obtaining the ground-truth uncertainties from real-world data can be costly or even infeasible. It is also known that commonly used uncertainty proxies in deep neural networks, such as the softmax output, tend to give overconfident estimates (Guo et al., 2017a). This overconfidence is further amplified under domain shifts, where the target (test) domain and the source training domain differ significantly. Such distributional shifts tend to aggravate the existing issues in uncertainty estimation, leading to wrong but overconfident predictions on unfamiliar samples (Li & Hoiem, 2020).

Many methods have been proposed to calibrate the confidence of deep learning models so that the uncertainty level of a model prediction reflects the likelihood of the true event (Guo et al., 2017a). Label smoothing is a popular approach to reduce overconfidence and to promote more uniform outputs (Szegedy et al., 2016). Temperature scaling is another method where the logit scores are rescaled by a calibrated temperature (Platt et al., 1999). Approaches such as Monte-Carlo sampling (Gal & Ghahramani, 2016) and Bayesian inference (Blundell et al., 2015; Riquelme et al., 2018) model uncertainties from a Bayesian perspective but are computationally more expensive. Even though these methods lead to more calibrated uncertainties, recent studies show that their results cannot be fully trusted under domain shift (Snoek et al., 2019).

**Our approach.** To handle domain shifts, we characterize the "overlap" between the source training data and the test data. Intuitively, if the test sample is highly unlikely in the training distribution,

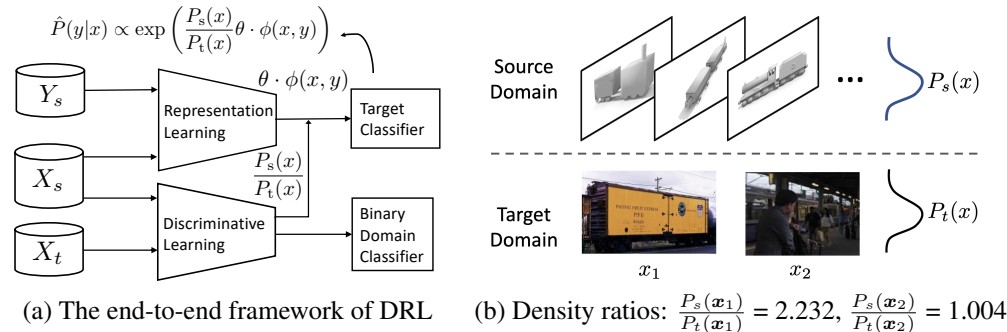

(a) The end-to-end framework of DRL

(b) Density ratios: $\frac{P_s(\boldsymbol{x}_1)}{P_t(\boldsymbol{x}_1)} = 2.232$, $\frac{P_s(\boldsymbol{x}_2)}{P_t(\boldsymbol{x}_2)} = 1.004$

Figure 1: (a) Architecture for end-to-end training of the proposed DRL framework (training details in Sec. 2.3). (b) Example images for category 'Train' in VisDA. The estimated density ratios for the easy and hard target images are shown, respectively. The DRL framework gives higher uncertain predictions for the harder example ($\boldsymbol{x}_2$) that is more cluttered and hence not well-represented in the source domain.

then the resulting confidence levels should be lowered. We incorporate this idea through the density ratio between the two domains and employ it for confidence calibration.

To make our idea more concrete, recall that the probability output of a standard neural network for classification can be expressed as $P(y|\boldsymbol{x}) \propto \exp\left(\boldsymbol{\theta}_y \cdot \phi(\boldsymbol{x})\right)$, where $\boldsymbol{x}$ is the predictor input, $\phi(\boldsymbol{x})$ is the data feature, and $\boldsymbol{\theta}_y$ is the model parameter of the $y$-th class. Instead, we propose the following predictive form for our neural network:

$$P(y|\boldsymbol{x}) \propto \exp\left(\frac{P_s(\boldsymbol{x})}{P_t(\boldsymbol{x})}\boldsymbol{\theta}_y \cdot \phi(\boldsymbol{x})\right), \tag{1}$$

where $P_s(\boldsymbol{x})$ and $P_t(\boldsymbol{x})$ are the densities of a data sample under the source and target distributions, respectively. When a target sample is close to the source domain (large $P_s(\boldsymbol{x})/P_t(\boldsymbol{x})$), the prediction is confident. However, when a target sample $\boldsymbol{x}$ is far away from the source distribution (small $P_s(\boldsymbol{x})/P_t(\boldsymbol{x})$), the confidence is lowered and the prediction is closer to a uniform distribution. This intuition is analogous to incorporating a sample-wise temperature to adjust the confidence according to the closeness of a test sample to the training distribution.

Eq. (1) is based on the distributionally robust learning (DRL) framework. DRL is an adversarial risk minimization framework that involves a two-player minimax game between a predictor and an adversary (Grünwald et al., 2004). While many previous DRL methods (Liu et al., 2020; Nakka et al., 2020) operate in low dimensional spaces using kernel density estimators for the density ratio estimation, we develop a DRL method to scale to real-world computer vision tasks, which is able to produce calibrated uncertainties under domain shift. **Our contributions are**:

**1)** We propose a DRL method for uncertainty estimation under domain shift. We introduce a binary domain classifier network which learns to predict the density ratios between source and target domains (See Fig. 1(a)). The domain classifier and the target classifier are trained simultaneously in an end-to-end fashion. We also introduce a regularized DRL framework to further promote smoothed model prediction and improve the calibration.

**2)** We show that the estimated density ratio reflects the distance of a test sample from both training and test distributions (Fig. 1(b)). Our experiments further empirically show that these estimates are also correlated with human selection frequency, based on the available ground-truth labels in ImageNetV2 (Recht et al., 2019a), which can be regarded as a proxy of human uncertainty perception.

**3)** We empirically show that the top-1 class predictions of DRL are more calibrated than the empirical risk minimization (ERM) and the temperature scaling on Office31 (Saenko et al., 2010), Office-Home (Venkateswara et al., 2017), and VisDA (Peng et al., 2017). We measure the level of calibration using expected calibration error (ECE), Brier Score and reliability plots.

**4)** We integrate our method as a plug-in module in downstream applications such as unsupervised domain adaption and semi-supervised learning, leading to significant improvements. For example, incorporating self-training (Zou et al., 2019) with DRL leads to state-of-the-art performance on VisDA-2017 (Peng et al., 2017) and a 6% improvement on hard examples. Incorporating Fix-

match (Sohn et al., 2020) with DRL improves the original Fixmatch by a relative 17% increase in accuracy under the cross-domain setting.

## 2 DISTRIBUTIONALLY ROBUST LEARNING

In this section, we first review the preliminaries of DRL (Sec. 2.1), followed by a proposed variant of DRL with class regularization (Sec. 2.2). We then propose an instantiation of DRL with a differentiable density ratio estimation network (Sec. 2.3). Finally, we show the application of DRL in UDA and SSL tasks (Sec. 2.4).

### 2.1 PRELIMINARIES

**Notations and definitions:** We denote the input and labels by random variables $X$ and $Y$. We also use $\boldsymbol{x} \in \mathbb{R}^d$ and $\mathcal{X}$ to represent the realization and sample space of $X$. Our goal is to find a predictor

$$\boldsymbol{F} : \mathbb{R}^d \mapsto \mathbb{R}^C \triangleq \{\boldsymbol{f}(\boldsymbol{x}) | \boldsymbol{x} \in \mathcal{X}, \boldsymbol{f}(\boldsymbol{x}) \in \mathbb{R}^C \cap \Delta\} \tag{2}$$

close to the true underlying $P_t(Y|X)$. Here, $d$, $C$ and $\Delta$ denote the input dimension, class number, and probabilistic simplex, respectively. We consider the problem with labeled data sampled from a source distribution $P_s(X, Y)$ and unlabeled data sampled from a target distribution $P_t(X)$, and use $eP_s(X, Y)$ to represent the empirical source distribution. In this work, we consider an important form of domain shift with the covariate shift assumption $P_s(X) \neq P_t(X)$, $P_s(Y|X) = P_t(Y|X)$.

**Motivation:** Traditional empirical risk minimization (ERM) frameworks tend to fail under covariate shift since ERM empirically learns a predictor $\hat{P}_s(Y|X)$ from the finite source data that usually fails to generalize to the target distribution. DRL was previously proposed to overcome this issue. DRL can be formulated as a two-player adversarial risk minimization game (Grünwald et al., 2004) with the predictor player minimizing a loss, while the adversary player maximizing the loss. The adversary is allowed to perturb the labels, subject to certain feature-matching constraints to ensure data-compatibility.

**Formulation:** Under covariate shift, DRL (Liu & Ziebart, 2014) deals with the mismatch between the expected loss and the training data, and is defined on the **target** data distribution:

$$\hat{P}_t(Y|X) = \underset{\boldsymbol{F}}{\operatorname{argmin}} \max_{\boldsymbol{G} \in \Sigma} \mathbb{E}_{\boldsymbol{x} \sim P_t(X)} \mathcal{L}\left(\boldsymbol{f}(\boldsymbol{x}), \boldsymbol{g}(\boldsymbol{x})\right), \tag{3}$$

where $\boldsymbol{f}(\boldsymbol{x}), \boldsymbol{g}(\boldsymbol{x}) \in \mathbb{R}^C$ are the conditional label distributions given an input $\boldsymbol{x}$ and $\boldsymbol{F}$, $\boldsymbol{G}$ are the entire distribution over all the input. $\mathbb{E}_{x \sim P_t} \mathcal{L}(\cdot)$ is an expected log loss on the **target** input defined:

$$\mathbb{E}_{\boldsymbol{x} \sim P_t(X)} \mathcal{L}(\boldsymbol{f}(\boldsymbol{x}), \boldsymbol{g}(\boldsymbol{x})) \triangleq \mathbb{E}_{\boldsymbol{x} \sim P_t(X)}[-\boldsymbol{g}(\boldsymbol{x}) \cdot \log \boldsymbol{f}(\boldsymbol{x})]. \tag{4}$$

$\boldsymbol{F}$ is the predictor player minimizing the loss function while $\boldsymbol{G}$ is the adversary maximizing the loss function. After solving this game, $\boldsymbol{F}$ is our estimate of $\hat{P}_t(Y|X)$, which we will use for the classification on the target domain.

Eq. 3 is defined on the **target** domain only. How could a predictor be properly trained while there are no target labels available? The answer is that the adversary $\boldsymbol{G}$ is implicitly constrained by the **source** features. We use the following constraints to make sure that $\boldsymbol{G}$ is close to $P_s(Y|X)$:

$$\Sigma = \{\boldsymbol{G} | \textstyle\sum_i g_y \phi(\boldsymbol{x}_i) = \sum_i \mathbb{I}[y_i = y] \phi(\boldsymbol{x}_i), \forall y\}, \tag{5}$$

where $\boldsymbol{x}_i \sim P_s(X)$ and $g_y$ is the $y$-th dimension of $\boldsymbol{g}$. Eq. 5 is a necessary but not sufficient condition for $\boldsymbol{G} = P_s(Y|X)$, thus serving as an implicit constraint for $\boldsymbol{G}$ to be close to the true $P_t(Y|X)$ under the covariate shift assumption. Given a predefined feature function $\phi$, when the adversary perturbs the conditional label distribution, certain aggregate function of $\phi$ on $\boldsymbol{g}$ should equal to the counterpart on the empirical source data.

**From DRL to density ratio:** When using the expected target logloss in Eq. 3, Eq. 1 is derived by solving the predictor $\boldsymbol{F}$. Here we refer the derivation details to (Liu & Ziebart, 2014) but emphasize the important properties of the prediction: representation-level conservativeness.

**Representation-level conservativeness:** The predictions have higher certainty for inputs closer to the source domain when $P_s(\boldsymbol{x})/P_t(\boldsymbol{x})$ is large. On the contrary, when $P_s(\boldsymbol{x})/P_t(\boldsymbol{x})$ is small, the prediction is more uncertain. This property reflects the model's ability to convey information about what it does not know through the model uncertainty.

## 2.2 CLASS-REGULARIZED DISTRIBUTIONALLY ROBUST LEARNING

Inspired by label smoothing (Szegedy et al., 2016) and regularization of neural network outputs (Pereyra et al., 2017), we can further add class-regularization to the conservative predictions in Eq. 1. Instead of doing it in a post-hoc way, we study how to incorporate class-regularization into the DRL framework. We propose to use a weighted logloss to penalize high confidence in the adversary's label prediction:

$$\hat{P}_t(Y|X) = \operatorname*{argmin}_{\boldsymbol{F}} \ \max_{\boldsymbol{G} \in \Sigma} \ \mathbb{E}_{\boldsymbol{x} \sim P_t(X)}[-\boldsymbol{g}(\boldsymbol{x}) \cdot \log \boldsymbol{f}(\boldsymbol{x})] - r\mathbb{E}_{\boldsymbol{x} \sim P_t(X)}\left[\boldsymbol{y} \odot \boldsymbol{g}(\boldsymbol{x}) \cdot \log \boldsymbol{f}(\boldsymbol{x})\right] \quad (6)$$

where $\boldsymbol{y}$ is the one-hot class vector, $\odot$ is the element-wise product, and $r \in [0, 1]$ is a hyper-parameter that controls the level of regularization. $\Sigma$ here is the same as in 3. We observe the whole loss function is a convex-concave function in terms of $\boldsymbol{f}$ and $\boldsymbol{g}$. According to the strong duality, we switch the order of the $\min$ and $\max$. With a fixed $\boldsymbol{g}$, $\boldsymbol{f} = \boldsymbol{g}$ is the optimal solution of the inner $\min$ problem. So we have the following lemma (we refer all the proofs to appendix Sec. A):

**Lemma 1.** *Eq. 6 can be reduced to a regularized maximum entropy problem with the estimator constrained:*

$$\max_{\boldsymbol{F} \in \Sigma} \ \mathbb{E}_{\boldsymbol{x} \sim P_t(X)}[-\boldsymbol{f}(\boldsymbol{x}) \cdot \log \boldsymbol{f}(\boldsymbol{x})] - r\mathbb{E}_{\boldsymbol{x} \sim P_t(X)}\left[\boldsymbol{y} \odot \boldsymbol{f}(\boldsymbol{x}) \cdot \log \boldsymbol{f}(x)\right] \quad (7)$$

*where $\Sigma$ is the same as Eq. 5, meaning $\boldsymbol{F}$ should be close to the empirical source $P_s(Y|X)$.*

**Theorem 1.** *The solution of Eq. 7 for takes the form: $\boldsymbol{f}_{\theta,r}(y|\boldsymbol{x}) \propto \exp\left(\frac{\frac{P_s(\boldsymbol{x})}{P_t(\boldsymbol{x})}\boldsymbol{\theta}_y \cdot \boldsymbol{\phi}(\boldsymbol{x}) + r\mathbb{I}(y)}{r\mathbb{I}(y)+1}\right)$,*

*where $\boldsymbol{\theta}$ represents the model parameters and $\mathbb{I}(y)$ is the yth dimension of the one-hot encoding $\boldsymbol{y}$.*

The proof of Theorem 1 follows the same principles when Eq. 1 is derived from Eq. 3. The training process of this model also follows the same procedure in (Liu & Ziebart, 2014), except that we use Theorem 1 to compute the prediction results. In training, we can compute the gradients using source labels where $\boldsymbol{y}$ is the one-hot encoding of each class. In testing, we set $\boldsymbol{y}$ to be an all-one vector.

**Class-level regularization:** Hyper-parameter $r$ adjusts the smoothness of $\boldsymbol{g}$'s label prediction in Eq. 6. It translates to the $r\boldsymbol{y}$ terms in the prediction form. Intuitively, this regularization increases the correct label's prediction result when it is small and decreases it when it is large, using a threshold of 1 to discriminate between 'large' and 'small'. This provides additional regularization and smoothness to the conservative prediction.

## 2.3 DIFFERENTIABLE DENSITY RATIO ESTIMATION

Estimating $P_s(\boldsymbol{x})/P_t(\boldsymbol{x})$ (Sugiyama et al., 2012) can be challenging, especially in high-dimensional spaces. Usually a plug-in estimator is used. But the plug-in estimates are usually sub-optimal for downstream tasks due to the different objectives. We propose an end-to-end training procedure for DRL such that **the density ratio estimator is trained together with the target classifier**. The key insight here is that we use two neural networks for classification and differentiating the two domains, respectively. See Fig. 1(a). We now first introduce differentiable density ratio estimation before introducing joint training loss and the parameter learning of the two networks.

**Differentiable density ratio estimation:** Based on the Bayes' rule, $P_s(\boldsymbol{x})/P_t(\boldsymbol{x})$ can be computed from a conditional domain classifier (Bickel et al., 2007): $\frac{P_t(\boldsymbol{x})}{P_s(\boldsymbol{x})} = \frac{P(\boldsymbol{x}|t)}{P(\boldsymbol{x}|s)} = \frac{P(t|\boldsymbol{x})P(s)}{P(s|\boldsymbol{x})P(t)}$. Concretely speaking, they can be estimated through binary classification using unlabeled source and target data with $\frac{P(s)}{P(t)}$ as a constant relating to number of source and target samples. On the other hand, we observe that $P_s(\boldsymbol{x})/P_t(\boldsymbol{x})$ can be a trainable weight for each sample, and can receive gradients from the training objective of DRL. Therefore, we propose to train a discriminative neural network to differentiate the two domains, which receives training signals from both the target classification loss in DRL and the binary classification loss. The weights trained this way lost their original properties as density ratios, but still reflects the relation between the two domains.

**Joint training loss:** Assume $\boldsymbol{\phi}(\boldsymbol{x}, \boldsymbol{w}_r)$ is the representation learning neural network with parameter $\boldsymbol{w}_r$. We further define $P_d(X)$ as the joint distribution of both source data and target data with their domain labels $D = \{\boldsymbol{d}(\boldsymbol{x})\}$. We denote $\boldsymbol{\tau}(\boldsymbol{x}, \boldsymbol{w}_d) = (\tau_s, \tau_t)$ (where $\tau_s + \tau_t = 1$) as the probability output of the source and target domains from a domain classifier with parameter $\boldsymbol{w}_d$. Our joint

training loss is defined as:

$$\min_{\boldsymbol{w}_r, \theta, \boldsymbol{w_d}} \mathbb{E}_{\boldsymbol{x} \sim P_t(X)} \left[ -\boldsymbol{g}_t(\boldsymbol{x}) \cdot \log \boldsymbol{f}(\boldsymbol{x}; \boldsymbol{w}_r, \theta, \boldsymbol{w}_d) \right] + \mathbb{E}_{\boldsymbol{x} \sim P_d(X)} \left[ -\boldsymbol{d}(\boldsymbol{x}) \cdot \log \boldsymbol{\tau}(\boldsymbol{x}, \boldsymbol{w}_d) \right] \quad (8)$$

where $\boldsymbol{g}_t(\boldsymbol{x}) = P_t(Y | X = \boldsymbol{x})$ and $\boldsymbol{f}(\boldsymbol{x}; \boldsymbol{w}_r, \theta, \boldsymbol{r}_d)$ takes the form in Theorem 1:

$$\boldsymbol{f}(\boldsymbol{x}; \boldsymbol{w}_r, \theta, \boldsymbol{r}_d) \propto \exp \left( \left( \frac{\tau_s(\boldsymbol{x}, \boldsymbol{w}_d)}{\tau_t(\boldsymbol{x}, \boldsymbol{w}_d)} \theta \cdot \phi(\boldsymbol{x}, \boldsymbol{w}_r) + r\mathbb{I}(y) \right) / (r\mathbb{I}(y) + 1) \right) \quad (9)$$

**Parameter learning**: Note that $\boldsymbol{g}_t$ in the first loss term of Eq. (8) concerns the conditional label distribution on target which is assumed to be not available. However, with the DRL formulation, we can compute and evaluate the gradients of $\theta$ and $\phi(\boldsymbol{x}, \boldsymbol{w}_r)$ directly with the change of measure in the derivation of the gradients (Liu & Ziebart, 2014), since the gradients are only associated with the source data and labels. We show this in Fig. 1(a) that the representation learning network $\phi$ only has source data as the input. We then update $\theta$ and $\boldsymbol{w}_r$ using the computed gradients and also directly back-propagate from the second loss term to update $\boldsymbol{w}_d$. Finally, we treat the densities as trainable variables and derive gradients for them from the first loss term (details shown in appendix Sec. B). By the Bayes rule, $\frac{P_s(\boldsymbol{x})}{P_t(\boldsymbol{x})} = \frac{\tau_s P(t)}{\tau_t P(s)}$, where $P(t)$ and $P(s)$ are the amount of unlabeled data from each of the domain during the training process. Since we usually use the same amount of source and target data in each batch, they cancel out by following $\frac{P(s)}{P(t)} = 1$. Then $\boldsymbol{f}$ in Theorem 1 is reduced to Eq. 9. Therefore, besides the binary classification loss, the parameter $\boldsymbol{w}_d$ of the discriminative network is also trained with gradients from the first loss term. Algorithm 1 summarizes the procedure.

---

**Algorithm 1** End-to-end Training for DRL

---

1: **Input**: Source data, target data, DNN $\phi$, DNN $\tau$, SGD optimizer SGD$_1$, SGD$_2$, learning rate $\gamma_1$ and $\gamma_2$, epoch number $T$.
2: **Initialization**: $\phi, \tau \leftarrow$ random initialization, epoch $\leftarrow 0$
3: **While** epoch $< T$
4:     **For** each mini-batch
5:         Update $\tau$ by SGD$_1(\gamma_1)$ using the combined gradients from both loss terms;
6:         Compute $\boldsymbol{f}$ using $\theta, \boldsymbol{w}_r$, and $\boldsymbol{w}_d$;
7:         Update $\phi, \theta$ by SGD$_2(\gamma_2)$ using derived gradients;
8:     epoch $\leftarrow$ epoch $+1$
9: **Output**: Trained networks $\phi, \tau$.

---

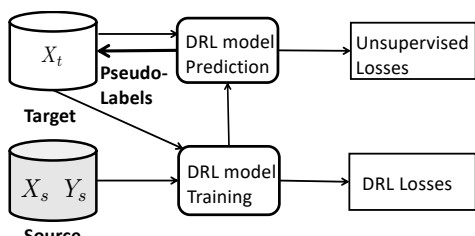

Figure 2: Formulation of the pseudo label based UDA or SSL methods with DRL. The unsupervised losses represent the loss function user can impose on the unlabeled target data. DRST conducts this procedure for multiple iterations, while DRSSL minimizes the unsupervised losses on the augmented target data.

## 2.4 APPLICATIONS TO UDA AND SSL

We show how to incorporate end-to-end DRL within a framework that takes unlabeled data for training. We introduce the general setting with two examples: self-training UDA and cross-domain SSL, followed by a new self-training algorithm DRST and a new SSL algorithm DRSSL.

**General settings:** In many cases, the source domain where models are trained may have abundant labels but the target domain lacks enough labels. Typical problems under this setting include unsupervised domain adaptation and (Zou et al., 2019) and semi-supervised learning (Sohn et al., 2020). In both cases, a common strategy is to treat the prediction results on the target data as pseudo labels to train the model on target data. In these methods, model confidence (softmax output) is often leveraged as the proxy to rank the reliability of pseudo labels, with the underlying assumption that there exists a positive correlation between model confidence and pseudo label quality. However, such an assumption requires accurate uncertainty estimation to avoid false usage of wrong pseudo labels which may poison the training. To this end, we incorporate DRL into these frameworks with to provide calibrated uncertainties. An illustration of this setting is shown in Figure 2.

**Distributionally robust self-training:** In UDA, we are given labeled source data and unlabeled target data and aim to achieve adaptation from the source to the target domain. Self-training is an effective method for UDA (Zou et al., 2019), where the training procedure in Fig. 2 is conducted multiple times. Here we propose DRST to plug the class-regularized DRL model into self-training.

The idea is to regard each training epoch as a new domain shift problem in DRL. After each training epoch, we make predictions on the target domain and select more confident data and generate pseudo-labels for them to merge into the source training data. Then the labeled source data and the newly pseudo-labeled target data become the new source set for the next training epoch.

**Distributionally robust semi-supervised learning:** In cross-domain SSL, we have few labeled source data and many unlabeled target data. We aim to utilize unlabeled data in the target domain to help representation learning and save the effort needed for labeling. One simple and effective strategy is to use pseudo-labels generated from weakly-augmented data to supervise strongly-augmented data (Sohn et al., 2020). Here 'weakly' means simple flip-and-shift data augmentation while 'strongly' follows the same strategy with (Sohn et al., 2020). Using DRL's prediction, we propose DRSSL, which assigns pseudo-labels more conservatively. We plug DRL into Fixmatch (Sohn et al., 2020). Here the unsupervised loss in Fig. 2 is the 'consistency loss' in (Sohn et al., 2020): $\mathcal{L}_u = \frac{1}{M} \sum_{m=1}^{M} \mathbb{I}(\max(\hat{P}(y_t^w | x_t^w)) > \tau) H(\hat{y}_t^w, \hat{P}(y_t^s | x_t^s))$, where $x_t^w$ and $y_t^w$ represent the weakly-augmented target data, $x_t^s$ and $y_t^s$ represent the strongly-augmented version of the same image data, and $\tau$ is a threshold for generating pseudo-labels $\hat{y}_t^w$.

## 3 EXPERIMENTS

We evaluate our method on benchmark datasets. DRL is evaluated as a method providing more calibrated uncertainties (Sec. 3.1), DRST as an effective UDA method (Sec. 3.2) and DRSSL as a cross-domain SSL method. We show additional results and details in appendix Sec. C, D.

**Datasets and methods:** We use Office31 (Saenko et al., 2010), Office-Home (Venkateswara et al., 2017) and VisDA (Peng et al., 2017) for evaluating DRL's uncertainties. We compare DRL with temperature scaling (TS), VADA (Shu et al., 2018) and source-only. We also train models using ImageNet (Deng et al., 2009) as the source domain and ImageNetV2 (Recht et al., 2019a) as the target domain to check the relationship between our estimated weights and the human selection frequencies (HSF). VisDA2017 is used to evaluate DRST, for which we compare with (1) traditional UDA baselines: MMD (Long et al., 2015), MCD (Saito et al., 2018b) and ADR (Saito et al., 2018a); (2) recent self-training UDA baselines: CBST (Zou et al., 2018) and CRST (Zou et al., 2019); (3) other uncertainty quantification or UDA methods combined with self-training baselines: AVH (Chen et al., 2020a) + CBST and DeepCORAL (Sun & Saenko, 2016)+CBST. In addition, we use CIFAR10, STL10 (Coates et al., 2011), MNIST (Lecun & Bottou, 1998) and SVHN (svh, 2011) to construct settings with few source labeled data and much unlabeled target data and show DRSSL's advantages in cross-domain SSL over Fixmatch (Sohn et al., 2020).

**Evaluation metrics:** Apart from accuracy, we also use Brier score (Brier, 1950), expected calibration error (ECE) (Guo et al., 2017a), and reliability plot (Dawid, 1982; Guo et al., 2017a; Naeini et al., 2015) to evaluate the performance of our proposed method and the baselines. Brier score measures the mean squared difference between the predicted probability assigned to the possible outcome and the actual outcome. Despite potential prob-

Table 1: ECE on ImageNetV2.

| HSF | Source | Temp.Scal. | DRL (Ours) |
|---|---|---|---|
| [0.0, 0.2] | 0.2694 | 0.2624 | **0.0129** |
| [0.2, 0.4] | 0.1818 | 0.1745 | **0.0036** |
| [0.4, 0.6] | 0.1344 | 0.1281 | **0.0012** |
| [0.6, 0.8] | 0.0667 | 0.0601 | **0.0019** |
| [0.8, 1.0] | 0.0319 | 0.0246 | **0.0019** |

lems with ECE (Nixon et al., 2019), it is still the most prevalent metric for the top-1 prediction. ECE is defined as the sum of average difference between prediction accuracy and confidence of different confidence bins (we use 15 bins in practice). For both the Brier score and ECE, the lower the score, the better the model is calibrated.

**Experiment setup:** For Office31 and Office-Home tasks, we use ResNet50 (He et al., 2016) as the backbone for all models to make the comparison fair. We train by SGD for 100 epochs and set the learning rate to 0.001. For VisDA, we use the ResNet101 (He et al., 2016) backbone and also the SGD optimizer. During the 20 epochs of training, the initial learning rate is set as $10^{-5}$ and the weight decay parameter is set as $5 \times 10^{-4}$. For ImageNet, we follow the standard training process of AlexNet (Krizhevsky et al., 2012) and VGG-19 (Simonyan & Zisserman, 2014), where the initial learning rate is 0.01 and we decay the learning rate by a factor of 10 every 30 epochs.

Figure 3: Density ratios vs HSF on ImageNet V2.

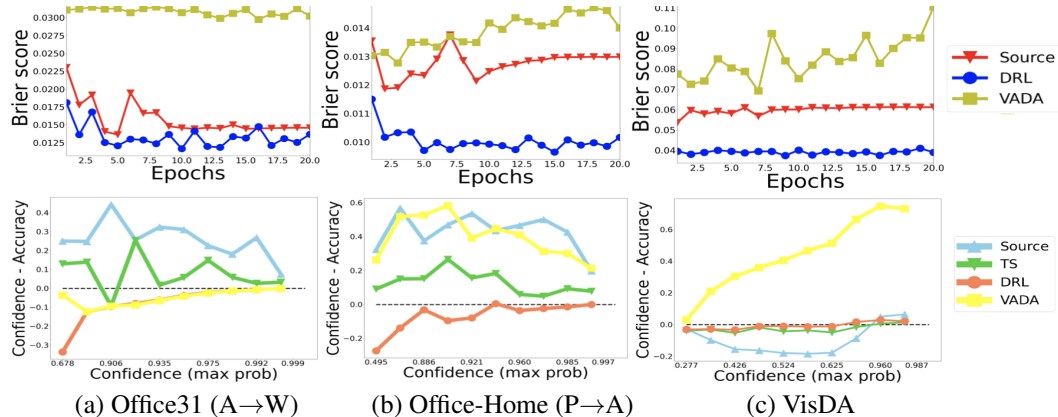

(a) Office31 (A→W)  (b) Office-Home (P→A)  (c) VisDA

Figure 4: Brier score (upper) and reliability diagrams (lower) on **Office31**, **Office-Home** and **VisDA**. DRL generates more calibrated uncertainties than source-only and temperature scaling and VADA Shu et al. (2018). Brier score measures the mean squared difference between the predicted probability and the actual outcome. For a fully calibrated classifier, the confidence should match the accuracy across the full range of confidence. Thus the closer the lines are to the dashed line, the more calibrated the method is. Our method gets more and more calibrated as confidence increases.

## 3.1 CALIBRATED UNCERTAINTIES FROM DRL

**DRL's calibrated confidence:** Fig. 4 demonstrates our more calibrated uncertainty. DRL tends to be underconfident and conservative but still stays closer to the calibration line (dashed line). Table 1 also shows our better ECE on ImageNetV2 in different human selection frequency (HSF) bins. Our additional results in appendix Sec. C show that the accuracy of DRL is also competitive.

**Density ratio v.s. human selection frequency:** Fig. 3 shows that the estimated weight is positively correlated with the human selection frequency (HSF) in ImageNetV2 (Recht et al., 2019b) under different neural network architectures. Here we regard $[0, 0.2]$ as low HSF and $(0.2, 1]$ as high HSF. Images with low HSF have smaller weights, indicating that they are visually harder for humans.

## 3.2 UNSUPERVISED DOMAIN ADAPTATION

**Initialization with ASG:** The quality of the source model has a significant impact on the final performance. The recently proposed automated synthetic-to-real generalization (ASG) model using ImageNet pretrained knowledge can be used to improve synthetic training (Chen et al., 2020b). Initializing with ASG trained model leads to significant improvements in self-training. Hence, we use the ASG pretrained model as an initialization, with results shown in Table 2's last four lines.

**Accuracy and calibration:** Fig. 5(a) and Table 2 show that DRST performs best in accuracy. Our vanilla version of DRST outperforms CRST by over 5% with ASG initialization. We improve the SOTA self-training accuracy on VisDA by over 1%. In the rest of the results, we use DRST to represent DRST-ASG. Fig. 5(b) shows that DRST also achieves higher calibrated confidence.

Table 2: Accuracy comparison with different methods on VisDA2017. "Skate" denotes "Skateboard".

| Method | Aero | Bike | Bus | Car | Horse | Knife | Motor | Person | Plant | Skate | Train | Truck | Mean |
|---|---|---|---|---|---|---|---|---|---|---|---|---|---|
| Source (Saito et al., 2018a) | 55.1 | 53.3 | 61.9 | 59.1 | 80.6 | 17.9 | 79.7 | 31.2 | 81.0 | 26.5 | 73.5 | 8.5 | 52.4 |
| MMD (Long et al., 2015) | 87.1 | 63.0 | 76.5 | 42.0 | 90.3 | 42.9 | 85.9 | 53.1 | 49.7 | 36.3 | 85.8 | 20.7 | 61.1 |
| MCD (Saito et al., 2018b) | 87.0 | 60.9 | **83.7** | 64.0 | 88.9 | 79.6 | 84.7 | 76.9 | 88.6 | 40.3 | 83.0 | 25.8 | 71.9 |
| ADR (Saito et al., 2018a) | 87.8 | 79.5 | **83.7** | 65.3 | 92.3 | 61.8 | 88.9 | 73.2 | 87.8 | 60.0 | 85.5 | 32.3 | 74.8 |
| CBST (Zou et al., 2018) | 87.2 | 78.8 | 56.5 | 55.4 | 85.1 | 79.2 | 83.8 | 77.7 | 82.8 | 88.8 | 69.0 | **72.0** | 76.4 |
| CRST (Zou et al., 2019) | 88.0 | 79.2 | 61.0 | 60.0 | 87.5 | 81.4 | 86.3 | 78.8 | 85.6 | 86.6 | 73.9 | 68.8 | 78.1 |
| CBST-AVH (Chen et al., 2020a) | 93.3 | 80.2 | 78.9 | 60.9 | 88.4 | 89.7 | 88.9 | 79.6 | 89.5 | 86.8 | 81.5 | 60.0 | 81.5 |
| CBST-DCORAL (Sun & Saenko, 2016) | 92.1 | 78.9 | 83.0 | 73.6 | 93.2 | 94.7 | 89.0 | 83.0 | 89.8 | 81.2 | 85.5 | 44.9 | 82.4 |
| **DRST** (ours) | 93.47 | 86.30 | 65.74 | 68.03 | 93.99 | 95.08 | 87.34 | 83.30 | 92.97 | 88.65 | 83.66 | 66.42 | 83.75 |
| ASG (Chen et al., 2020b) | 88.81 | 68.55 | 65.31 | 78.06 | **95.78** | 9.11 | 84.89 | 29.58 | 82.13 | 33.76 | **86.00** | 12.04 | 61.17 |
| CBST-ASG (Chen et al., 2020b) | **95.12** | **86.53** | 79.83 | 76.01 | 94.61 | 92.34 | 85.94 | 75.08 | 89.23 | 82.16 | 73.42 | 56.49 | 82.23 |
| CRST-ASG (Chen et al., 2020b) | 92.38 | 81.30 | 74.63 | **84.40** | 90.90 | 92.43 | **91.65** | **83.78** | **94.92** | 88.12 | 74.88 | 61.10 | 84.21 |
| **DRST-ASG** (ours) | 94.51 | 85.58 | 76.50 | 77.18 | 94.39 | **95.33** | 88.89 | 81.23 | 94.22 | **90.36** | 81.75 | 63.10 | **85.25** |

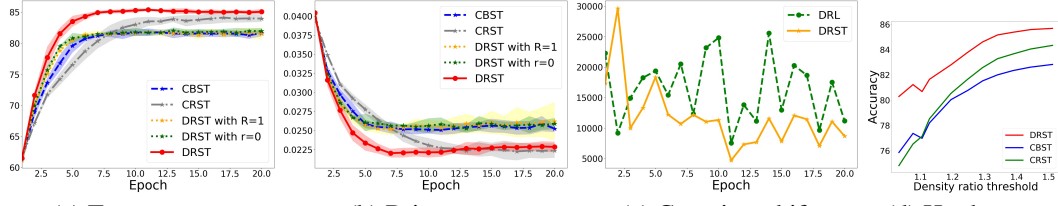

| (a) Test accuracy | (b) Brier score | (c) Covariate shift | (d) Harder cases |

Figure 5: (a)-(b) Main results on VisDA-17 (5 random seeds) with the test accuracy and Brier score. The results show that DRST outperforms the baselines significantly. (c) Distribution gap $P_s(\phi(\boldsymbol{x}))/P_t(\phi(\boldsymbol{x})) - P_s(\phi(\boldsymbol{x}), y)/P_t(\phi(\boldsymbol{x}), y)$ as a proxy of covariate shift. DRST helps to further reduce this gap over DRL with self-training. (d) Accuracy vs. estimated weights. The improvement from DRST increases on harder examples (lower weights).

**Ablation study:** Fig. 5(a)(b) include two ablation methods. In the first ablation, we set $r$ to 0 so that there is no class regularization in DRL ("r = 0"). The prediction then follows the form in Eq. (1). In the second ablation, we set the weights to 1 to mute the differentiable density ratio estimation in our method so that there is no representation level conservativeness ("R = 1"). DRST achieves the best performance when both components are present.

**Covariate shift:** Fig. 5(c) shows $P_s(\phi(\boldsymbol{x}))/P_t(\phi(\boldsymbol{x})) - P_s(\phi(\boldsymbol{x}), \boldsymbol{y})/P_t(\phi(\boldsymbol{x}), \boldsymbol{y})$ using discriminative density ratio estimators (per class). This serves as a proxy of covariate shift as it becomes 0 when covariate shift holds. We can see that the gap decreases with self-training, which shows that even though covariate shift may not hold in the beginning, self-training helps to promote this assumption with better aligned domains and more discriminative feature distributions.

**Improvement on hard examples:** Fig. 5 (d) demonstrates that DRST achieves larger performance gain compared to the baselines on target samples with smaller weights. Recall that data is not well-represented in the source domain when $P_s(\boldsymbol{x})/P_t(\boldsymbol{x})$ is small (Fig. 1(b)). Therefore, DRST provides robust performance on harder examples in the target domain.

**Density ratios:** In Fig. 1(b), a harder example obtains a lower density ratio due to its vague shape. Our density ratios reflect the closeness of a sample to the two domains. More examples are in appendix Sec. C. Moreover, the magnitude of our estimated density ratios are modest in general within the range $[0.1, 10]$. This is due to the regularization by the other network's learning signals.

**Improved attention:** Fig. 6 visualizes the network attention of DRST and the baselines using Grad-CAM (Selvaraju et al., 2017), where DRST renders improved attention with better object coverage.

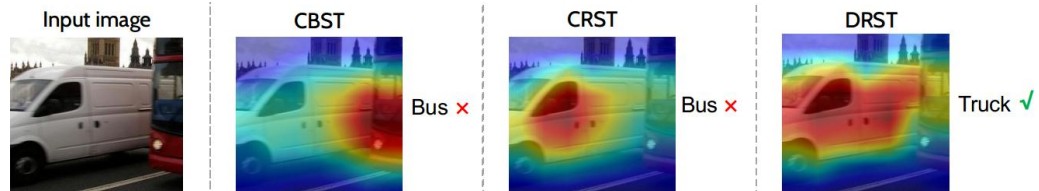

Figure 6: Model attention visualized using Grad-Cam (Selvaraju et al., 2017). We also show the predicted labels by different methods. Our method captures the shape features of the image better.

### 3.3 DISTRIBUTIONALLY ROBUST SEMI-SUPERVISED LEARNING

We use two pairs of source and target domains in this experiment: Source: CIFAR10 / Target: STL10 and Source: MNIST / Target: SVHN. Table 3 show that DRL-powered SSL method improves the original Fixmatch significantly in

Table 3: Accuracies for cross-domain SSL.

| Test set | CIFAR10 | STL10 | MNIST | SVHN |
|---|---|---|---|---|
| Fixmatch (Sohn et al., 2020) | 91.60 | 59.61 | 99.43 | 26.50 |
| **DRSSL** (Ours) | **95.17** | **69.38** | **99.46** | **30.96** |

cross-domain semi-supervised learning tasks. Note our setting is different from UDA where source labeled data is abundant. SSL focuses on using unlabeled augmented data to help learning from few labeled data. In the CIFAR10 to STL10 case, we only have 4k labeled source data. For MNIST to SVHN, we have 40k labeled source data. The results show that DRL is beneficial for generating high-quality pseudo supervision for unlabeled data under the cross-domain SSL setting.

## 4 RELATED WORK

**Distributionally robust learning:** Several different learning algorithms can be derived from this framework (Liu & Ziebart, 2014; 2017; Chen et al., 2016; Fathony et al., 2016; 2018). One limitation of this line of work is that it is only suitable for low-dimensional data. The other inconvenience is that it requires the density ratio to be estimated beforehand. Our work instead deals with high-dimensional data and estimates the density ratio along with the learning process. Another line of work perturbs the covariate variable (Ben-Tal et al., 2013; Shafieezadeh Abadeh et al., 2015; Sinha et al., 2018; Hu et al., 2018; Najafi et al., 2019) and focuses on model robustness against adversarial perturbations. In our work, we make DRL practical in high-dimensional vision tasks by introducing an end-to-end trained domain classifier.

**UDA and SSL:** There is a rich literature in UDA focusing on representation alignment between the source and target domains (Fernando et al., 2014; Pan et al., 2010; Sun et al., 2016; Wang et al., 2018; Baktashmotlagh et al., 2013; Na et al., 2021). It is also prevalent to use a discriminator network to differentiate two learned representations (adversarially) to locate such a feature space (Ganin et al., 2016; Tzeng et al., 2017; Ajakan et al., 2014; Long et al., 2018; Sankaranarayanan et al., 2018). Some other works focus on more specific shift assumptions, such as covariate shift (Shimodaira, 2000) and label shift (Lipton et al., 2018; Azizzadenesheli et al., 2019). For the covariate shift case, even though various density ratio estimation methods were investigated before (Sugiyama et al., 2012), only few works explore the possibility to apply them to high-dimensional data (Khan et al., 2019; Moustakides & Basioti, 2019; Park et al., 2020). Recently, multiple self-training methods have been proposed for UDA (Zou et al., 2018; 2019) and many theoretical understandings of self-training are also developed (Kumar et al., 2020; Chen et al., 2020c). Similarly, rich literature also exist in pseudo-labeling for SSL (Berthelot et al., 2019; Arazo et al., 2020; Miyato et al., 2018; Sohn et al., 2020). In this paper, we focus on utilizing DRL's model uncertainty to help with choosing better pseudo-labeled data from the target domain in UDA and cross-domain SSL.

**Uncertainty quantification with deep models:** Bayesian methods largely contribute to complementing deep learning with uncertainty quantification (Gal, 2016; Welling & Teh, 2011; Kingma & Welling, 2013; Louizos & Welling, 2017; Blundell et al., 2015; Riquelme et al., 2018; Gal & Ghahramani, 2015; Gal et al., 2017; Kendall & Gal, 2017; Mandt et al., 2017). However, with a large number of parameters, Bayesian deep learning may suffer from computational inefficiency. Other calibration methods include temperature scaling (Platt et al., 1999; Guo et al., 2017b), deep ensembles (Lakshminarayanan et al., 2017), calibration regression (Kuleshov et al., 2018), quantile regression (Romano et al., 2019) and trainable calibration measure (Kumar et al., 2018). These methods do not apply to our setting because we do not have access to the target labels. Recent work on calibrating or uncertainties under domain shift either only focus on "using" but not "generating" the uncertainties (Han et al., 2019; Lee & Lee, 2020; Kurmi et al., 2019) or focus on the importance weighting setup (Park et al., 2020; Wang et al., 2020). In this paper, we generate more calibrated uncertainties using the DRL framework, where the density ratios are inherent.

## 5 CONCLUSION AND DISCUSSION

In this paper, we have studied uncertainty estimation under distribution shift using the distributionally robust learning framework. We demonstrate that density estimation can be integrated into the learning process by using a domain classifier. We propose differentiable density ratio estimation and develop end-to-end training techniques for our proposed method. Using DRL's more calibrated model confidence helps to generate better pseudo-labels for self-training in UDA and cross-domain SSL. We also empirically demonstrate that the density ratios learned from our domain classifier reflect the hardness of an image, showing a positive correlation with the human selection frequencies. In the future, we are interested in relaxing the assumptions made and study different shifts in the DRL framework.

**Density ratio estimation:** Our density ratios are estimated from a differentiable domain classifier and is not guaranteed to match the true density ratios. Thus our density ratios should not be used in circumstances where true density ratios are absolutely required. However, we show that our density ratios do reflect the relation between the two distributions and benefit the downstream tasks.

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
