# OpenReview forum: "Distributionally Robust Learning for Uncertainty Calibration under Domain Shift"
_ICLR.cc/2022/Conference — ICLR 2022 Submitted_

### Official Review · Reviewer_6Mv8 · 2021-11-02

**Correctness:** 3
**Technical Novelty And Significance:** 2
**Empirical Novelty And Significance:** 2
**Recommendation:** 5
**Confidence:** 3

**Main Review:**

Strengths:

* The proposed use of density ratios makes an attempt to quantify the domain shift. This is an improvement over other works which take domain shift for a given.
* Uncertainty is evaluated using both ECE and Brier score. (Usually papers choose only one of the two.)
* Section 2.2 derives use of density ratios to scale (temperature of) uncertainty from theory in Distributionally Robust Learning, which is a novel combination of views. Subsequently, Figure 3 shows that the density ratios correspond to human interpretation of domain differences. This correspondence validates the design choice for density ratio.

Weaknesses:

* “Intuitively, if the test sample is highly unlikely in the training distribution, then the resulting confidence levels should be lowered”. I don’t agree with this intuition. Let’s say the training distribution is of synthetic renderings and the test distribution of real-life pictures, then confidence could still be high when the same object is depicted. What would be a counter-argument to this statement and how does that influence the results?

* The human selection scores seem to be a major motivation of this paper, but the figure misses error bars. When the ratio estimates change from 2.2 to 2.6, what is the noise in these numbers? (In other words, how statistically significant is this observation?)

* The calibration and training of density ratios is not well explained. Densities are known to be hard to interpret in high dimensions [3]. Could the relevance of these density ratios be shown via some experiment?

* Question: From figure 4 it seems that DRL generally shows signs of underconfidence (C-A negative), while TS and VADA signs of overconfidence (C-A positive). Could this effect be explained?


Minor comments,
  * Section 3: why use 5 bins for ECE? A lower number of bins introduces more bias, especially for high accuracy models (where the last bin is most occupied) [4]. This makes the ECE results less reliable.
  * Section 2.2: This section has many connections to the research in GANs, where a similar density ratio is estimated [1,2].

NB: These are meant as suggestions and do not influence the review. Use of papers on arXiv is not part of the review procedure.

[1] Goodfellow et al. "Generative adversarial nets." NeurIPS 2014

[2] Arjovsky et al. Wasserstein generative adversarial networks ICML 2017

[3] Nalisnick, et al. "Detecting out-of-distribution inputs to deep generative models using a test for typicality." arXiv 2019

[4] Roelofs, et al. "Mitigating bias in calibration error estimation." arXiv 2020


**Summary Of The Paper:**

This paper proposes scaled uncertainty prediction in the context of unsupervised domain adaptation. The problem in unsupervised domain adaptation is to obtain predictions in a target domain where no (or few) labels are available. A secondary problem then is the calibration of predictive uncertainties in the target domain. This paper proposes a model of uncertainty proportional to the (log) of density ratios between domains. Results include ECE and Brier scores on two common Domain Adaptation datasets.

**Summary Of The Review:**

This paper derives a new scaling for predictive uncertainty in unsupervised domain adaptation. Motivated by Distributionally Robust Learning theory, the scaling, density ratios, are verified against human frequency scores. Results show better calibration under similar (slightly better) predictive performance.

---

> ### Author Response · Authors · 2021-11-12
> **Response to Reviewer  6Mv8**
>
> Thank you for the thorough comments and detailed questions. We are encouraged by your recognition of our contributions, including method's novelty and improvement over other works. Due to space limit, we will separate the response into two pages, where the second page addresses the minor comments. We now respond to your concerns point-by-point:
>
> **Regarding the doubt on the intuition and proposed statement:**
>
> There is some misunderstanding here. This sentence’s context is the original DRL method. In our work, if the shape representation is captured by a deep neural network, for example, the object in real life with the same shape would be considered to be close to the synthetic training distribution. By “highly unlikely”, we mean cases like: training distribution as the synthetic rendering of a horse looking from the front, and testing distribution as the real-life picture of a horse looking from the back. Under this situation, when tested on the target domain, it is hard to differentiate whether the image is a horse or a cattle looking from the back. Thus lowering the confidence level is a reasonable thing to do. As mentioned in the review, there is a gap between the real-world vision tasks and the original DRL’s intuition. This gap is exactly what we want to close by proposing deep learning to learn the latent representation in the DRL framework. We will clarify this intuition in our revision.
>
> **Regarding the missing error bars for Figure 3:**
>
> We have provided a new plot with error bars added. The new plot can be both referred to in the new version of the paper as well as in this anonymous [link](https://drive.google.com/file/d/1AflWQgP3XEpdxY5-FJgTTl7nLb0sp71F/view?usp=sharing).
>
> **Regarding the relevance between density ratios and calibration:**
>
> We agree with the reviewer that the density ratios are hard to interpret. Due to the fact that we do not have the ground truth, we refer to some implicit metrics. In Figure 3, we look at the relation between human visual hardness (HSF) and density ratio. We also showed examples of “hard samples” with their corresponding density ratios and HSF in Figure 1(b), appendix Figure 10, appendix Figure 11 and appendix Figure 14.
>
> We also mention in our conclusion that our density ratio is not guaranteed to match the true density ratio but still reflects the relation between the two distributions and benefits the downstream tasks.
>
> In terms of how the density ratio affects the calibration, we have ablation studies with density ratio R = 1 in UDA (Figure 5(b)) that shows our model’s Brier score benefits from training density ratios. Figure 4 also has a comparison between DRL and source-only (density ratio all equal to 1), that demonstrates better reliability plots. We also want to note that calibration is usually a model-level metric that is usually evaluated using the whole dataset while density ratios are instance-level measurements. Therefore, it’s a bit hard to do a controlled experiment to test the effects of the sample-wise density ratios.
>
> Here we added two experiments on the VisDA2017 dataset to show the relationship between density ratio and confidence as well as accuracy.The results are in this [link](https://drive.google.com/file/d/1nWpQytizuwTmESjqzuUSr95WVGfV4TnA/view?usp=sharing). Fig A shows the correlation between density ratio and predicted confidence. Fig B shows the correlation between density ratio and (confidence-accuracy). The evaluation is done on the VisDA2017 test dataset using our DRL model.
>
> From Fig A, we can see that the density ratios are positively correlated with confidence for the prediction results ,which means that smaller density ratios give less confident prediction results. For Fig B, (confidence-accuracy) is a measure used by the reliability plot that indicates some degree of calibration. We can observe that we are more calibrated for samples that are closer to the source domain (high density ratio).
>
> **Regarding the observation of DRL’s underconfidence:**
>
> DRL considers the worst-case model that fits the data, so it is motivated from a pessimistic perspective. But pessimism would be the right thing for confidence when the shift is large. We demonstrate that even when the model is underconfident, it is more calibrated. Moreover, this uncertainty from underconfidence benefits the downstream tasks.

---

> ### Author Response · Authors · 2021-11-12
> **Response to Reviewer 6Mv8 (cont'd)**
>
> Thank you for spending time reading through the long response. The following response is for answering the other questions mentioned:
>
> **Regarding the bin number used for calculating ECE:**
>
> Thanks for pointing this out. We realized that it is a typo here, we actually used 15 bins. We will fix this in our revision. Please refer to our code (in fig_plot.py of this [repository](https://anonymous.4open.science/r/Deep-Distributionally-Robust-Learning-for-Calibrated-Uncertainties-under-Domain-Shift-7C2E)) to check out. Using bins of 15 is a common strategy [b] and we followed [a] for calculating ECE.
>
> **Regarding connections with GAN:**
>
> Our density ratio is a term derived from the DRL which depicts the likelihood a sample belongs to source or target domain. This discriminative idea looks similar to GAN, but is actually rather different because we do not have any generative components and the density ratio is not directly estimated in GAN as well. Our discriminator component is also not only updated by discrimination loss but also by the classification loss.
>
> [a] Chuan Guo, Geoff Pleiss, Yu Sun, and Kilian Q Weinberger. On calibration of modern neural networks. In ICML, 2017a.
>
> [b] Zhizhong Li and Derek Hoiem. Improving confidence estimates for unfamiliar examples. In CVPR, 2020.

---

### Official Review · Reviewer_xXg4 · 2021-11-02

**Correctness:** 3
**Technical Novelty And Significance:** 2
**Empirical Novelty And Significance:** 2
**Recommendation:** 5
**Confidence:** 3

**Main Review:**

1. The paper studies an interesting and important research question: calibrate uncertainties under domain shift. And the authors proposed a novel method based on DRL and density ratio estimation. The proposed method can be seen as a plug-in tool for existing methods.
2. The paper may overclaim in the abstract that the authors "consider the case where the source distribution differs significantly from the target distribution". The proposed method has a covariate shift assumption. Then how could the two distributions differ SIGNIFICANTLY? Does the proposed method still work when the support of the source and target domain are (largely) different?
3. Density ratio estimation is known to be hard when the input data is complex/in high-dimension. I have some doubts about how well the density ratio is estimated in the proposed method. I think further investigation in the learned density ratios is necessary to understand how the proposed method works. Figure 3 shows a coarse study on this point but I'm not convinced by this simple result. The discussions and ablation studies may not enough to show this point. Also, the threshold of low/high HSF seems quite artificial.
4. The proposed method is based on a strong covariate shift assumption, which is very likely to be violated in real-world applications. The authors show that self-training may help to relax this assumption. But how about the proposed method without self-training? Self-training is already a strong additive in weakly-supervised learning for achieving good performance. Perhaps it is the self-training part in DRST that makes the proposed method work? I hope to see more results for the proposed method without self-training.

**Summary Of The Paper:**

In the paper, the authors propose a framework for learning calibrated uncertainties under domain shift based on a distributionally robust learning (DRL) framework. As a plug-in module, the proposed method benefits downstream tasks of unsupervised domain adaptation and semi-supervised learning in experiments.


**Summary Of The Review:**

The research question is important and useful but the writing of the paper may overclaim the usage of the proposed method. Also, it may be hard to conclude how the proposed method works under the current evidence.

---

> ### Author Response · Authors · 2021-11-12
> **Response to Reviewer xXg4:**
>
> Thank you for the time and detailed reviews. We appreciate the acknowledgement in the importance of the research direction and the method's novelty. Below is the response to your concerns:
>
> **“Overclaim” and conditions where the method may work:**
>
> We understand the concern about overclaiming and we realize this misinterpretation is due to the word choice of “significantly”. We will clarify and modify the wording in the revision. One misunderstanding we want to clarify is, the covariate shift assumption does cover those cases when $P_s(x)$ and $P_t(x)$ differ significantly even when $P(y|x)$ keeps the same. Here, “$x$” can be regarded as the learned representation from neural networks, but not the raw features in pixels. We do study cases when they are very different, for example, VisDA2017 is a sim2real problem where the source domain (GTA5, synthetic data) and target domain (Cityscapes, real-world images) are very different.
>
> To conclude, our method still works well when the source and target domains are very different. Indeed, our method is motivated by a theoretical framework which makes a covariate shift assumption. And we followed this framework as a general principle in designing our method (such as Eq. 5 in the main paper). However, this does not imply that our method has to rely on very strict covariate shift assumption in order to perform well.
>
> When the two domains differ significantly, the density ratio $P_s(x)/P_t(x)$ generally will be very small, and our method will generate more uncertain predictions and in the extreme case, the predicted probabilities follow a uniform distribution. As shown in our experiments, our uncertainty estimate is more calibrated and benefits downstream tasks that are heavily dependent on the uncertainties.
>
> **Deeper insight into density ratio:**
>
> We agree with the reviewer that density ratio estimation is hard, as well as the evaluation of it. In our case, we do not have the ground truth and can only rely on implicit metrics. Apart from Figure 3, we showed examples of “hard samples” with their corresponding density ratios in Figure 1(b), density ratios with HSF in appendix Figure 10, appendix Figure 11 and appendix Figure 14. We also have ablation studies with density ratio $R = 1$ in UDA (Figure 5 (a)(b)) that shows our model benefits from training density ratios.
>
> Also, we added two experiments on the VisDA2017 dataset to show the relationship between density ratio and confidence as well as accuracy. The results are in this [link] (https://drive.google.com/file/d/1nWpQytizuwTmESjqzuUSr95WVGfV4TnA/view?usp=sharing). Fig A shows the correlation between density ratio and predicted confidence. Fig B shows the correlation between density ratio and (confidence-accuracy). The evaluation is done on the VisDA2017 test dataset using our DRL model.
>
> From Fig A, we can see that the density ratios are positively correlated with confidence for the prediction results ,which means that smaller density ratios give less confident prediction results. For Fig B, (confidence-accuracy) is a measure used by the reliability plot that indicates some degree of calibration. We can observe that we are more calibrated for samples that are closer to the source domain (high density ratio).
>
> Finally, we also want to mention (this is also in our conclusion section) that our density ratio is not guaranteed to match the true density ratio but still reflects the relation between the two distributions and benefits the downstream tasks.
>
> **Performance without self-training:**
>
> The table below presents results without self-training:
>
> |  | Source | DRL-Source |
> | :---: | :---: | :---: |
> | Accuracy (%) | 52.4 | 65.11 | 61.17 | 65.20|
> | |  ASG  | DRL-ASG |
> | Accuracy (%) | 61.17 | 65.20|
>
> Source is ResNet101 and ASG is another source model. DRL-xxx means that DRL is initialized with source model xxx. We can observe a great boost in accuracy when adopting DRL on different source models.
>
> Also, as shown in Figure 4, we compared DRL with source-only models, TS and VADA to justify its performance in uncertainty calibration.
> To conclude, we are not claiming the strong predictive performance for DRL itself, but with its main advantage in improving uncertainty estimation, DRL is able to perform as a plug-in module to boost source models’ performance, as well as self-training methods’ performance.

---

> > ### Comment · Reviewer_xXg4 · 2021-11-26
> > **Post-rebuttal comments**
> >
> > Thank the authors very much for the rebuttal! After reading the rebuttal, the authors have addressed most of my concerns and conducted additional experiments. I will increase my score from 3 to 5: marginally below the acceptance threshold. I still have some concerns about the claimed "our density ratio is not guaranteed to match the true density ratio but still reflects the relation between the two distributions", and not very clear of the details of how to conduct differentiable density ratio estimation (i.e., how this differs from prior work).
> > To improve the paper, I think the authors may
> > 1) properly address how much distribution shift could the proposed method handle
> > 2) make the differentiable density ratio estimation and joint training parts more clear and emphasize the relationship of prior work and the proposed method on this point
> > 3) add the additional experimental results (i.e., performance without self-training) to the paper

---

> > > ### Author Response · Authors · 2021-11-26
> > > **Further response**
> > >
> > > Thank you for raising the score. We will first address the additional questions and then present our plan to further improve the paper following your comments.
> > >
> > > **Regarding how much distribution shift:** We want to further clarify that even though the original DRL method is based on a covariate shift assumption, our method does not require that. Please refer to our general response. We will also clarify this in our further paper revision.
> > >
> > > **Regarding the differentiable density ratio estimation:** We understand that some of the details can be elaborated in the paper. We want to clarify that our method does not conduct the density ratio estimation task independently, but the density ratio estimation network is trained along with our target classification network. In our experiment, we demonstrate that our estimated density ratios do reveal samples’ relative “distance” from the two domains. Thus we say “our density ratio is not guaranteed to match the true density ratio" (since it is trained with the additional purpose to help the target classification task), "but still reflects the relation between the two distributions”.
> > >
> > > **Regarding how differentiable density ratio estimation is different from the previous work:** We give a more detailed version as follows:
> > >
> > > 1. Construct a binary classifier to classify a sample as source or target (same as prior methods);
> > > 2. Do division on the predicted binary output to achieve P_s(x)/P_t(x) (different from prior works because the density ratio is in the form of P_s(x)/P_t(x) due to the DRL formulation, while it is usually P_t(x)/P_s(x) in importance sampling literature);
> > > 3. Update the binary classifier using the binary cross-entropy loss and the classification loss, as shown in Eq. 8. Here the classification loss is the loss for the target classification task, say 1000-class classification for ImageNet (different from prior methods);
> > > 4. When updating the parameters using classification loss, the gradients are calculated separately for P_s(x) and P_t(x) according to their derivation in the appendix (different from prior methods).
> > >
> > > To further improve the paper, we will include the above clarification in the paper and also add the additional experimental results to the paper. We hope the above helps resolve your concerns. If there are any additional questions, we are more than willing to answer them.

---

### Official Review · Reviewer_zkq9 · 2021-11-04

**Correctness:** 3
**Technical Novelty And Significance:** 2
**Empirical Novelty And Significance:** 2
**Recommendation:** 5
**Confidence:** 5

**Main Review:**

Strengths:
+ The authors incorporate the density ratio estimation method into the existing distributional robust learning (DRL) framework and successfully achieve a better calibration under domain shift.
+ They further integrate their method as a plug-in module in downstream applications such as unsupervised domain adaption and semi-supervised learning, leading to significant improvements. This part is new to me and riveting. I appreciate the authors for not only making the predictions more calibrated but also achieving higher performance.
+ Competitive performance on various tasks including Office31, Office-Home, and VisDA-2017 demonstrate the superiority of DRL over empirical risk minimization (ERM) and the temperature scaling method measured by expected calibration error (ECE), Brier Score and reliability plots.
+ This paper is well-written and easy to follow.

Weaknesses:
+ My first major concern is the novelty of this paper. There are mainly two technical parts of this paper: the density ratio estimation method and distributional robust learning (DRL). However, density ratio estimation by a domain classifier is common practice in domain adaptation and transfer learning. Moreover, this technique has already been successfully tailored into calibration under domain shift [1][2]. As for the distributional robust learning (DRL) framework, it has been successfully proposed in Liu & Ziebart, 2014 as stated by the authors.
The differences between these related works with the proposed method should be fully discussed to make the contribution enough for this top-tier conference.
+ Another major concern is the performance of FixMatch on CIFAR10. The reported number (91.60) is much lower than that reported in the original paper of FixMatch, which is about 95.7?
+ What is the definition of human selection frequencies (HSF)? Why should the estimated weight match HSF? I highly encourage the authors to deeply analysis this topic to bring more insights.
+ Why Brier Score of Temperature Scaling (TS) is missing in Figure 4?

[1] S. Park, O. Bastani, J. Weimer, and I. Lee. Calibrated prediction with covariate shift via unsupervised domain adaptation. 2020.

[2] X.Wang, M. Long, J. Wang, and M. Jordan. Transferable Calibration with Lower Bias and Variance in Domain Adaptation. In NeurIPS, 2020


**Summary Of The Paper:**

This paper focuses on the problem of uncertainty calibration under distribution shift. By using a domain classifier in the distributional robust learning (DRL) framework, the authors estimate the density ratio between the source and target domain to achieve well-calibrated predictions under domain shift. A regularized DRL framework is further proposed to promote smoothed model prediction and improve the calibration. Experiments on Office31, Office-Home, and VisDA-2017 demonstrate the superiority of DRL over empirical risk minimization (ERM) and the temperature scaling method measured by expected calibration error (ECE), Brier Score, and reliability plots.

**Summary Of The Review:**

A uncertainty calibration method under domain shift with a distributionally robust learning framework that show competitive performance on comprehensive experiments but lacks enough novelty.

---

> ### Author Response · Authors · 2021-11-12
> **Response to Reviewer zkq9**
>
> Thanks for your recognition of our contributions and detailed comments as well as suggestions. In the following response, we will address your concerns on the method's novelty. Due to the character limit, we will use another response page to answer the other questions.
>
> **Regarding the paper’s novelty:**
>
> To resolve the concerns on the novelty of the paper, we restate our contribution with an explicit explanation about our relation with the original work in DRL, [1], [2], and the works in density ratio estimation. We will also add these discussions to the main paper.
>
> First, our method contains significant changes that make it more practical than the DRL framework proposed in (Liu & Ziebart, 2014). We 1) propose a new regularization form that can further promote smoothed prediction and improve the calibration performance (the resulting predictive form and the learning using new gradients are both novel); 2) enable the density ratio estimation network to be trained in an end-to-end fashion in the DRL framework (this requires a new training objective and a new learning mechanism, which is non-trivial). As a result, our work has shown promising results on high dimensional image data including Office31,OfficeHome, VisDA2017 and ImageNet, while the previous method (Liu & Ziebart, 2014) is applied on relatively low-dimensional or toy data (synthetic Gaussian data and UCI dataset); 3) evaluate the uncertainty estimates from DRL and also utilize them in UDA and SSL tasks, which would not be possible with the previous DRL method that requires a pre-estimated density ratio for each sample.
>
> Second, indeed density ratio estimation is a commonly used technique, but its incorporation into the DRL framework together with end-to-end deep learning induces new challenges and requires new methodologies in training. By making the density ratio a trainable parameter, we derived gradients from the classification loss for $P_s(x)$ and $P_t(x)$. See detailed derivation in Appendix Section B. Thus, the density ratio is supervised by both the domain classification loss and the DRL loss. To the best of our knowledge, this derivation and training techniques have not been studied or developed. This also introduces new interesting properties to the density ratio as the magnitude of the resulting ratio is always modest. Please refer to our discussion of density ratios in the experiment section.
>
> Finally, our method is very different from methods such as [1] and [2]. [1] proposes to use importance weighting (IW) to correct for the shift between source to target. Even though IW can also be regarded as a density ratio and can be trained using a domain classifier, it is derived from an empirical risk minimization (ERM) framework, while we work in the DRL framework. Note that the density ratio in DRL is a natural derivation from the DRL theory instead of an artificially designed parameter for neural network models. For more details about the differences between ERM and DRL, please refer to section 2.1. In addition, our density ratio appears in the softmax predictive form but not in the loss function, as in IW methods. Moreover, our density ratio ($P_s(x)/P_t(x)$) is the inverse of IW ($P_t(x)/P_s(x)$), which indicates that it has a significantly different role from that of IW. Our density ratio would adjust the predictions of samples to be uncertain when they are farther from source but closer to target, while the IW method would upweight such samples in training. [2] proposes the importance-weighted ECE as a new calibration metric under covariate shift and only focuses on post-processing the uncertainty estimates after training. Again, our density ratio is the inverse of IW and appears in the softmax predictive form. Our work derives novel predictive forms from DRL, trains the model in an end-to-end way and incorporates the calibrated uncertainty to benefit the down-stream tasks.

---

> > ### Comment · Reviewer_zkq9 · 2021-11-30
> > **Post-rebuttal comments**
> >
> > Thanks for providing such a detailed response. The authors addressed some of my concerns, except for the novelty part. I still think the contribution of this paper cannot meet the high bar of ICLR, though it may be a bit subjective. After reading reviews from other reviewers and taking a second look at the revised paper, I think this is a borderline paper. If other reviewers highly recommend accepting it, it will be ok to me.

---

> ### Author Response · Authors · 2021-11-12
> **Response to Reviewer zkq9 (cont'd)**
>
> Thank you for spending time reading through the long response. The following response is for answering the other questions mentioned:
>
> **Regarding reported FixMatch accuracy:**
>
> This is due to our different experiment setting from the original FixMatch paper (see our added explanation on the protocol details in Section D.5 in appendix). Under our setting, the training dataset is sampled from the whole dataset. Specifically, our training set is CIFAR10 and our testing set is STL10. However, the CIFAR10 dataset’s labels are not consistent with STL10, where the “frog” class in CIFAR10 is replaced by “monkey” in STL10. Thus, we eliminate these two classes. The task we do here is then CIFAR9 to STL9. This approach of preprocessing the dataset is also used in [a].The discussion of this procedure was included in Section D.5 in appendix.
>
> **Regarding HSF and estimated density ratio:**
>
> Thanks for pointing this out. “Match” is a bad word choice here. The right expression would be there is a “positive correlation” between HSF and estimated density ratios. We will fix it in the paper.
>
> We adopt the idea of HSF from [b] where HSF is defined as: the average number of times an image gets picked by a crowd of annotators from a pool belonging to a certain specified category. HSF can be seen as a representation for human visual hardness, where the lower HSF is, the harder for humans to classify an image to a certain category. Recall that estimated density ratio indicates whether a target image is close to  the clean source domain. The definition is $P_s(x)/P_t(x)$, thus the lower it is, the less likely the image $x$ is from the source data distribution and tends to be a hard sample (also shown in Figure 1(b)).
>
> Therefore, the positive correlation between the estimated weight and HSF indicates that our estimated weight gives our model correct information about the uncertainty of the sample.
>
> **Regarding the missing Brier Score in Figure 4:**
>
> The first row of Figure 4 intends to show how uncertainty metrics change during the training process. TS is usually performed and evaluated after the training is completed, thus applying TS on models that are not fully trained is not a common practice. However, we still present an additional figure including the TS method to provide deeper insights in this anonymous link: https://drive.google.com/file/d/1I2xnCWGhesJ9bZqtWsj2HdIWeZ2XSPr4/view?usp=sharing.
>
>
> [a] Rui Shu, Hung H Bui, Hirokazu Narui, and Stefano Ermon. A dirt-t approach to unsupervised domain adaptation. ICLR, 2018.
>
> [b] Beidi Chen, Weiyang Liu, Zhiding Yu, Jan Kautz, Anshumali Shrivastava, Animesh Garg, and Anima Anandkumar. Angular visual hardness. In ICML, 2020a.

---

> ### Author Response · Authors · 2021-11-30
> **Further response**
>
> Thank you again for the reviews. If you have any further concerns, feel free to post them and we will reply in a timely manner.

---

### Official Review · Reviewer_St2k · 2021-11-08

**Correctness:** 3
**Technical Novelty And Significance:** 2
**Empirical Novelty And Significance:** 2
**Recommendation:** 6
**Confidence:** 4

**Main Review:**

The paper proposes an uncertainty ( density ratio) of source-target prediction to adapt for the target domain. The novelty is somewhat limited; other works also tackled the distributional shift using the confidence and prediction of domain classifiers.[d][f]. The paper is lacking in some of these aspects:
Clarifications:
What is the $\sum$ in Eq 3
How to present work different/related adversarial domain adaptation methods; this discussion will clarify the paper further.  using the Bayes theorem, the Differentiable density ratio estimation can be treated as the discriminator's prediction.( discussed in sec 2.3)
Do source and target predictors share some parameters?
In Eq 6and  7, how to obtain y for unsupervised domain adaptation case. Is it a class label or domain label?
Performance:
The performance is not compared with recent state-of-the-art methods in domain adaption[a][b][c][d][f] in real workd dataset such as VisDA.
Related comparison:
The paper missed some of the useful references that tackle the domain adaptation using uncertainty.[d][e][f]. I suggest authors to compare their methods with these related works in terms of performance.


[a] FixBi: Bridging Domain Spaces for Unsupervised Domain Adaptation, CVPR 2021
[b]d-SNE: Domain Adaptation using Stochastic Neighborhood Embedding, CVPR, 2019
[c]Self-adaptive Re-weighted Adversarial Domain Adaptation, IJCAI 2020
[d]Unsupervised Domain Adaptation via Calibrating Uncertainties, CVPR workshop 2019
[e]Model Uncertainty for Unsupervised Domain Adaptation ICIP 2019
[f] Attending to Discriminative Certainty for Domain Adaptation, CVPR 2019

**Summary Of The Paper:**

The paper proposes an uncertainty ( density ratio) of source-target prediction to adapt for the target domain. The methods also validated in the robust semi-supervised tasks.

**Summary Of The Review:**

The paper is lacking in terms of performance on domain adaptation tasks as compared to state-of-the-art methods. Some more clarification about the method will increase the readability of the paper.

---

> ### Author Response · Authors · 2021-11-12
> **Response to Reviewer St2k**
>
> Thank you for the time and detailed suggestions. We now respond to your concerns:
>
> **Regarding the meaning of $\Sigma$ in Eq 3:**
> $\Sigma$ is the feature matching constraint that guarantees the adversary $G$ to have the feature expectation close to the training data, see Eq. 5.
>
> **Regarding discussion of different adversarial domain adaptation methods:**
> We cited several adversarial DA work in the related work section but did not go into detail, due to the fact that many adversarial domain adaptation methods generally focus on the performance (say accuracy) of the UDA tasks but ignore the predictive uncertainty and calibration of the prediction results. We provide a brief discussion here and will briefly discuss this in our revision:
> DANN[a] uses a similar framework as ours, which also includes a category classifier and a domain classifier. However, the domain classifier only aims at aligning the source and target distributions and the category classifier is driven by the cross-entropy loss. VADA[b] is another method (which is discussed and compared with in our paper) that introduces a new penalty term and uses a teacher-student learning framework. Other works include [c][d][e]. These DA methods are motivated by the idea of finding a common representation of the source and target domain through adversarial learning, while our method aims to improve the performance by calibrating the prediction results to provide better supervision and avoid overconfidence on uncertain samples.
>
> [a] Ganin Y, Ustinova E, Ajakan H, et al. Domain-adversarial training of neural networks[J]. The journal of machine learning research, 2016, 17(1): 2096-2030.
>
> [b] Takeru Miyato, Shin-ichi Maeda, Masanori Koyama, and Shin Ishii. Virtual adversarial training: a regularization method for supervised and semi-supervised learning. IEEE Trans. PAMI, 2018.
>
> [c] Mingsheng Long, Zhangjie Cao, Jianmin Wang, and Michael I Jordan. Conditional adversarial domain adaptation. In NeurIPS, 2018.
>
> [d] Eric Tzeng, Judy Hoffman, Kate Saenko, and Trevor Darrell. Adversarial discriminative domain adaptation. In CVPR, 2017.
>
> [e] Swami Sankaranarayanan, Yogesh Balaji, Carlos D Castillo, and Rama Chellappa. Generate to adapt: Aligning domains using generative adversarial networks. In CVPR, 2018.
>
> **Regarding the parameters of source and target predictors:**
>
> Yes they do share the same parameters. The predictor’s parameters are the same when used to predict on source data and target data. This is also a common practice for most domain adaptation methods.
>
>
> **Regarding $y$ in Eq 6 and 7:**
>
> The $y$ in Eq 6 and 7 is the known class label in the source domain or the predicted pseudo-label in the target domain. Under the unsupervised domain adaptation setting, source data, source label and target data are available during the training. Concretely speaking, $y$ is the class label for the corresponding samples, and is replaced by the model’s prediction result when ground-truth is not available.
>
> **Regarding the performance comparison with mentioned references:**
>
> We thank the reviewer for providing the references. We will add them to the related work section. In addition, we provide the following discussion on the comparison between our method the the mentioned work:
> Our method focuses on generating a calibrated uncertainty estimate under domain shift while [a][b][c] do not involve uncertainty estimation. [d][e][f] are more closely related to our work. [d] employs variational Bayes learning for uncertainty estimation to calibrate the predictions while our method is based on the DRL framework. [d] also achieves an accuracy of 80.59±1.39% on VisDA2017 while ours is 83.75% using the same source model. Moreover, [d] does not analyze the calibration of prediction results. [e] aims to minimize the model uncertainty to learn domain-invariant representations, while our DRL method adjusts the uncertainty using the density ratios. [e] yields a result of 78.5% on VisDA2017, which is even lower than ours.  [f] focuses on the domain classifier’s classification certainty and uses it to guide the attention of representation learning. In the following table, we provide a comparison of our method with one version of [f] on the two most difficult tasks in Office31. Our method achieves competitive performance:
>
> |  Task   | D$\rightarrow$A  | W$\rightarrow$A |
> |  ----  | ----  | --- |
> | CADA-W[f]  |  68.9% | 68.3% |
> | DRST  | 69.2% | 68.3% |

---

> ### Author Response · Authors · 2021-11-30
> **Further response**
>
> Thank you again for the reviews. If you have any further concerns, feel free to post them and we will reply in a timely manner.

---

### Author Response · Authors · 2021-11-12
**General Response**

Dear area chair and reviewers,
We appreciate the reviewers' time and valuable comments. Overall, the reviewers think that our work is well-motivated (R2, R3), well-written (R2), novel (R3, R4), and provides comprehensive experiment results (R2, R4).  The major concerns lie in our comparison and relation with some related work (R1), technical novelty (R2) and deeper insight into the method as well as the estimated density ratios (R3, R4).

However, we believe there exists misunderstandings regarding our work, which led the reviewers to incorrectly position our work with existing ones (R1, R2) and interpret our motivations (R3, R4).

For R1 and R2, we want to clarify that our work is the first to make the DRL framework practical in high-dimensional vision tasks. We provide not only novel extensions of the frameworks, novel derivations in the new end-to-end training techniques, but also novel analysis on the resulting uncertainty estimates and how it would benefit important downstream tasks.  We will extend our discussion on related works based on the mentioned UDA literature (R1), the previous DRL framework, as well as the density ratio estimation methods (R2).

For R3 and R4, we would like to clarify that though the method is based on statistical DRL under the covariate shift assumption, we study how the intuition can be successfully applied to real-world domain shift cases when representations are learned by deep neural networks. We are simply motivated by a theoretical framework which assumes covariate shift and our method does not rely on the covariate shift to perform well. Its performance is empirically verified in the paper on various real world datasets. As for the estimated density ratio, it is an important component but its quality is hard to evaluate. Thus we performed various experiments in the paper and provided additional evidence in the response to demonstrate its effectiveness and how it affects the model predictions. Please refer to our detailed answers to your questions in the individual response.

In the following individual responses, we provide detailed answers to all the questions/comments and supplement new experiment results to further strengthen our contributions.

We have also updated the paper in the following aspects: 1) added the references the reviewers suggested and updated the related work section; 2) updated Figure 3 with error bars; 3) fixed typos about the binning strategy of calculating ECE and updated the ECE results using 15 bins; 4) updated the abstract to make it more precise in describing the problem setting.

---

### Author Response · Authors · 2021-11-20
**Additional General Response**

Dear reviewers,

We believe we have resolved all the issues raised in the reviews. But we are more than willing to answer any further questions and make further updates to the paper. Please let us know if there are any and we will respond in a timely manner.

---

### Decision · Program_Chairs · 2022-01-20

**Decision:**

Reject

**Comment:**

This paper investigates the problem of uncertainty calibration under distribution shift. Based on a distributionally robust learning (DRL) framework, the paper estimates the density ratio between the source and target domains to achieve well-calibrated predictions under domain shift.
As a plug-in module, the proposed method benefits downstream tasks of unsupervised domain adaptation and semi-supervised learning in experiments on Office31, Office-Home, and VisDA-2017, demonstrating the superiority over empirical risk minimization (ERM) and the temperature scaling method measured by expected calibration error (ECE), Brier Score, and reliability plots.

After extensive interactions and discussions on the paper, the final scores were 6/5/5/5. AC considered the paper itself, and all reviews, author responses, and discussions, and reject the paper from the following concerns:
+ *Overclaimed Novelty*: This paper is mainly based on the well-established competitive distributionally robust learning (DRL) framework. The contribution of the newly-proposed regularization form that can further promote smoothed prediction and improve the calibration performance is relatively trivial. The designs of the resulting predictive form and the learning using new gradients, mentioned by the authors in the rebuttal, need further exploration and elaboration to verify its contributions.
+ *Lack of Clarifications*: Some key points mentioned by several reviewers are still not clear. For example, how well the density ratio is estimated, especially in high dimensions? Further, a positive correlation between HSF and density ratio is not enough to prove the main argument of the paper.
+ *Some statements are not well-supported*: For example, the statement that "the harder the examples are, the farther away the examples are from the source domain", claimed by the author in the rebuttal, is untenable.

In summary, this paper studies a promising research direction of uncertainty estimation, but the work cannot be accepted before addressing the reviewers' comments. I suggest the authors to substantially revise their work by incorporating all rebuttal material as well as addressing the remaining concerns.